# Assembly of the 81.6 Mb centromere of pea chromosome 6 elucidates the structure and evolution of metapolycentric chromosomes

Jiří Macas[1]*, Laura Ávila Robledillo[1], Jonathan Kreplak[2], Petr Novák[1], Andrea Koblížková[1], Iva Vrbová[1], Judith Burstin[2], Pavel Neumann[1]

1 Biology Centre, Czech Academy of Sciences, Institute of Plant Molecular Biology, Branišovská 31, České Budějovice, Czech Republic, 2 Agroécologie, AgroSup Dijon, INRA, Univ. Bourgogne, Univ. Bourgogne Franche-Comté, Dijon, France

* macas@umbr.cas.cz

**Data Availability Statement:** Raw data used for scaffolding, sequence assembly, and ChIP-seq

## Abstract

Centromeres in the legume genera *Pisum* and *Lathyrus* exhibit unique morphological characteristics, including extended primary constrictions and multiple separate domains of centromeric chromatin. These so-called metapolycentromeres resemble an intermediate form between monocentric and holocentric types, and therefore provide a great opportunity for studying the transitions between different types of centromere organizations. However, because of the exceedingly large and highly repetitive nature of metapolycentromeres, highly contiguous assemblies needed for these studies are lacking. Here, we report on the assembly and analysis of a 177.6 Mb region of pea (*Pisum sativum*) chromosome 6, including the 81.6 Mb centromere region (CEN6) and adjacent chromosome arms. Genes, DNA methylation profiles, and most of the repeats were uniformly distributed within the centromere, and their densities in CEN6 and chromosome arms were similar. The exception was an accumulation of satellite DNA in CEN6, where it formed multiple arrays up to 2 Mb in length. Centromeric chromatin, characterized by the presence of the CENH3 protein, was predominantly associated with arrays of three different satellite repeats; however, five other satellites present in CEN6 lacked CENH3. The presence of CENH3 chromatin was found to determine the spatial distribution of the respective satellites during the cell cycle. Finally, oligo-FISH painting experiments, performed using probes specifically designed to label the genomic regions corresponding to CEN6 in *Pisum*, *Lathyrus*, and *Vicia* species, revealed that metapolycentromeres evolved via the expansion of centromeric chromatin into neighboring chromosomal regions and the accumulation of novel satellite repeats. However, in some of these species, centromere evolution also involved chromosomal translocations and centromere repositioning.

## Author summary

Despite their conserved function, centromeres exhibit considerable variation in their morphology and sequence composition. For example, centromere activity is restricted to a single region in monocentric chromosomes, but is distributed along the entire chromosome length in holocentric chromosomes. The principles of centromere evolution that led to

analysis are available from the European Nucleotide Archive (https://www.ebi.ac.uk/ena/, study accession no. PRJEB54858). The final CEN6 sequence and its annotation are available from the Czech National Repository (https://doi.org/10.48700/datst.8t29q-nfr77) and from the interactive genome browser JBrowse (http://w3lamc.umbr.cas.cz/lamc/jbrowse.php).

**Funding:** This work was funded by the Czech Science Foundation grant no. GACR 20-24252S to J.M. The funders had no role in study design, data collection and analysis, decision to publish, or preparation of the manuscript.

**Competing interests:** The authors declare no competing interests.

this variation are largely unknown, partly due to the lack of high-quality centromere assemblies. Here, we present an assembly of the pea metapolycentromere, a unique type of centromere that represents an intermediate stage between monocentric and holocentric organizations. This study not only provides a detailed insight into sequence organization, but also reveals possible mechanisms for the formation of the metapolycentromere through the spread of centromeric chromatin and the accumulation of satellite DNA.

## Introduction

Centromeres are chromosomal regions that facilitate faithful chromosome segregation during cell division by serving as an anchor point for the assembly of the kinetochore, a protein complex that connects centromeric chromatin to spindle microtubules [1]. In most species, the position of the centromere on chromosomes is determined epigenetically by the presence of the centromere-specific histone variant CENH3 (also called CENP-A) and other proteins comprising the constitutive centromere-associated network [2]. Despite their conserved function, eukaryotic centromeres are highly variable in size, structure, and sequence composition, a phenomenon called the centromere paradox [3].

Centromeres exhibit two distinct types of organization, which influence the overall morphology of chromosomes [4]. They are either restricted to a single specific region that forms a primary constriction during mitosis (monocentric chromosomes) or distributed along the entire chromosome length (holocentric chromosomes). Species with monocentric chromosomes are more common and presumably ancestral. Several phylogenetic lineages of animals and plants have independently transitioned to holocentricity [5]. Recently, another type of centromere organization has been described in the legume genera *Pisum* and *Lathyrus* [6,7]. These species possess "metapolycentric" chromosomes characterized by extended primary constrictions, which account for up to one-third of the chromosome length in metaphase and contain multiple domains of centromeric chromatin characterized by the presence of CENH3. These CENH3 domains are located along the outer periphery of the primary constriction and interact with the mitotic spindle; however, the interior of the constriction consists of CENH3-free chromatin. This morphology, together with the distribution of certain histone phosphorylation marks [8] strongly resembles chromatin organization on holocentric chromosomes, suggesting that metapolycentric chromosomes may represent an intermediate state between monocentric and holocentric chromosomes [4,8]. Thus, metapolycentric chromosomes provide a unique opportunity for studying the changes associated with the transition between different centromere organizations.

The molecular and evolutionary mechanisms leading to centromere variation remain poorly understood, because of difficulties in sequencing and assembling centromeric regions [9]. Deciphering the complete nucleotide sequence of centromeres in plants is complicated by the large size of these genome regions and their accumulation of highly repetitive DNA sequences such as long-terminal repeat (LTR)-retrotransposons and satellite DNA (satDNA) [10]. In particular, satDNA is a major obstacle to the gapless assembly of centromeres because it is arranged in megabase-sized arrays of almost identical, tandemly arranged monomers. At the same time, satDNA is of particular interest because it is known to be a key sequence component that interacts with CENH3-containing nucleosomes in many centromeres [11].

Recent advances in sequencing, computational, and cytogenetic techniques have ushered in a new era of centromere research. In this regard, the so-called long-read sequencing technologies, which include the Pacific Biosciences (PacBio) and Oxford Nanopore Technologies

(ONT) platforms, have provided a real breakthrough by offering the ability to generate "ultra-long" reads that can efficiently resolve satellite repeats. The utility of these technologies, together with novel scaffolding and computational approaches specifically tailored to repeat-rich genomic regions, was best demonstrated by the completion of the gapless assembly of all human centromeres [12,13]. Complete centromere assemblies have also been recently reported for several species of higher plants, including maize (*Zea mays*) [14,15], Arabidopsis (*Arabidopsis thaliana*) [16,17], and rice (*Oryza sativa*) [18], while near-complete assemblies have been achieved in additional species such as tomato (*Solanum lycopersicum*) [19]. Despite these advances, the number of species with centromere assemblies is still very limited and does not reflect centromere variation in higher plants.

In this study, we constructed the centromere assembly of garden pea (*Pisum sativum* L. cv. Cameor), a species with metapolycentric chromosomes. In addition to their exceptional organization, the centromeres of pea are populated with a large number of different satellite repeats [6,20], which is in contrast to plant species studied previously, which showed only one or few satellites occupying the centromeres of all chromosomes. Although the first genome draft of the same pea genotype is available [21], it lacks most of the repeat-rich centromeric regions because of the inherent limitations of the short-read sequencing technology used to generate this assembly. To overcome this limitation, we used long-read sequencing technologies to generate new sequence data, which were assembled and verified using a combination of bioinformatics and cytogenetic approaches. We selected the centromere of pea chromosome 6 (CEN6) for this study because this chromosome has the largest primary constriction (estimated at 70–100 Mb) carrying multiple satellite repeats associated with CENH3 chromatin [6]. The assembly was used to address the following: (1) how CEN6 differs in sequence composition and long-range organization from its neighboring chromosome arms and from the centromeres of other plant species, (2) how the linear sequence of metapolycentromere transforms into the specific three-dimensional structure observed on pea metaphase chromosomes; and (3) whether metapolycentromeres arise from regional centromeres by spreading of CENH3 chromatin to neighboring chromosomal regions or by expansion due to the accumulation of repetitive DNA.

## Results

### Assembly of the metapolycentromere of pea chromosome 6

We performed long-read sequencing, together with extensive manual curation and assembly verification by cytogenetic mapping, to obtain a highly contiguous and reliable sequence of CEN6 (S1 Fig). First, we optimized the protocol for generating long nanopore reads from pea. This resulted in 119.6 Gb (27.8x coverage) of sequence data represented by reads ranging 30–801 kb in length (N50 = 83.8 kb). A portion of the ultralong reads (>120 kb, 8.5x coverage, N50 = 171.7 kb) were then used to create scaffolds, starting with reads containing single-copy marker sequences mapped cytogenetically or genetically to CEN6 or with reads containing CEN6-specific satellite repeats. These "seed" reads were gradually extended by repeated semi-automated identification of terminally overlapping ultralong reads in both directions until scaffolds from adjacent seeds were merged. This procedure was relatively laborious because of the manual curation involved, but it allowed us to obtain verified scaffolds free of structural misassemblies that often affect repeat-rich regions. In the next step, contigs generated from highly accurate PacBio HiFi reads (73.1 Gb; 17x coverage) using two alternative assemblers (HiCanu and Hifiasm) were compared with the nanopore scaffolds. With the exception of two missing duplications (306 kb and 5,243 kb), there were no large structural discrepancies between the HiFi contigs and the nanopore scaffolds, with identical long-range structures of

several satDNA arrays of up to 2 Mb in length. Moreover, some highly homogenized satDNA arrays that could not be scaffolded with nanopore reads were fully assembled from the HiFi reads. This result justified the use of HiFi contigs for scaffolding the remaining regions not covered by nanopore scaffolds (S1 Fig) and for using HiFi reads to polish the entire assembly. During and after the scaffolding process, the assembly was verified by multicolor fluorescence in situ hybridization (FISH) mapping of selected satellite repeats and single-copy markers on pea chromosome 6 at different levels of condensation (pachytene, prometaphase, and metaphase). This approach resulted in a 177,603,725 bp-long assembly of the entire CEN6 and its adjacent chromosomal regions, with only a single gap located in one of the FabTR-10 satellite arrays (Fig 1A and 1B).

## Structure and sequence composition of CEN6

The assembly was annotated with respect to all major types of genomic sequences, including genes, tandem repeats, and various groups of transposable elements. We also generated chromatin immunoprecipitation-sequencing (ChIP-seq) reads using antibodies for both variants of the pea CENH3 protein to analyze the distribution of centromeric chromatin along the CEN6 sequence. This revealed multiple distinct regions of CENH3 accumulation up to ~1 Mb in length (Fig 1C). Because the transition of primary constriction to chromosome arms on metaphase chromosome 6 is marked by the positions of the outermost CENH3 loci (Fig 1A), the positions of the first and last CENH3 peaks were used to define an 81.6 Mb region in the assembly corresponding to the primary constriction (Fig 1B). Mapping the molecular marker sequences from the pea genetic map [22] onto the assembly revealed that the annotated constriction overlapped with the nonrecombining region of the linkage group LGII, further confirming its correct placement in the assembly (Fig 2).

The locations showing the highest accumulation of CENH3, which appeared as peaks in the ChIP-seq analysis track, were always associated with satDNA arrays (Fig 1C and 1D). These arrays included FabTR-10 repeats, which were located at multiple positions in CEN6, and FabTR-48 and FabTR-49, each of which occupied only a single locus. By contrast, other large satellites in CEN6, such as FabTR-85, -106, and -107, with arrays up to 2 Mb in size, were free of CENH3. Pea contains two variants of the CENH3 protein that differ in sequence and can be distinguished with specific antibodies [8]. The use of these two antibodies in ChIP-seq experiments revealed that the distribution patterns of the two CENH3 variants were identical (S2 Fig).

The extended primary constriction showed no significant difference in sequence composition when compared with the adjacent assembly regions representing the proximal parts of the short and long arms of chromosome 6, except for the accumulation of satDNA (Fig 1E). LTR-retrotransposons, including the lineage of Ty3/gypsy Ogre elements, a dominant repeat in the pea genome, showed uniform distribution along the entire assembly. Similar distributions were exhibited by Ty1/copia elements and DNA transposons. The lineage of Ty3/gypsy CRM elements, known to target plant centromeres [23], was found partially enriched in the constriction; however, these elements occur in the pea genome only in hundreds of copies and therefore have no significant effect on centromere composition. Annotation of the centromeric DNA revealed 602 genes, which were supported by the RNA-seq data, indicating that these genes were transcriptionally active. The gene density in the centromere was 7.4/Mb (or 8.3/Mb, excluding regions with satDNA arrays), which was lower than that in the adjacent chromosome arms (12.0/Mb).

Since the tools for analyzing DNA methylation in nanopore reads have recently become available [24], we examined the frequencies of cytosine methylation in all three contexts known from higher plants. DNA methylation profiles were generally similar between the

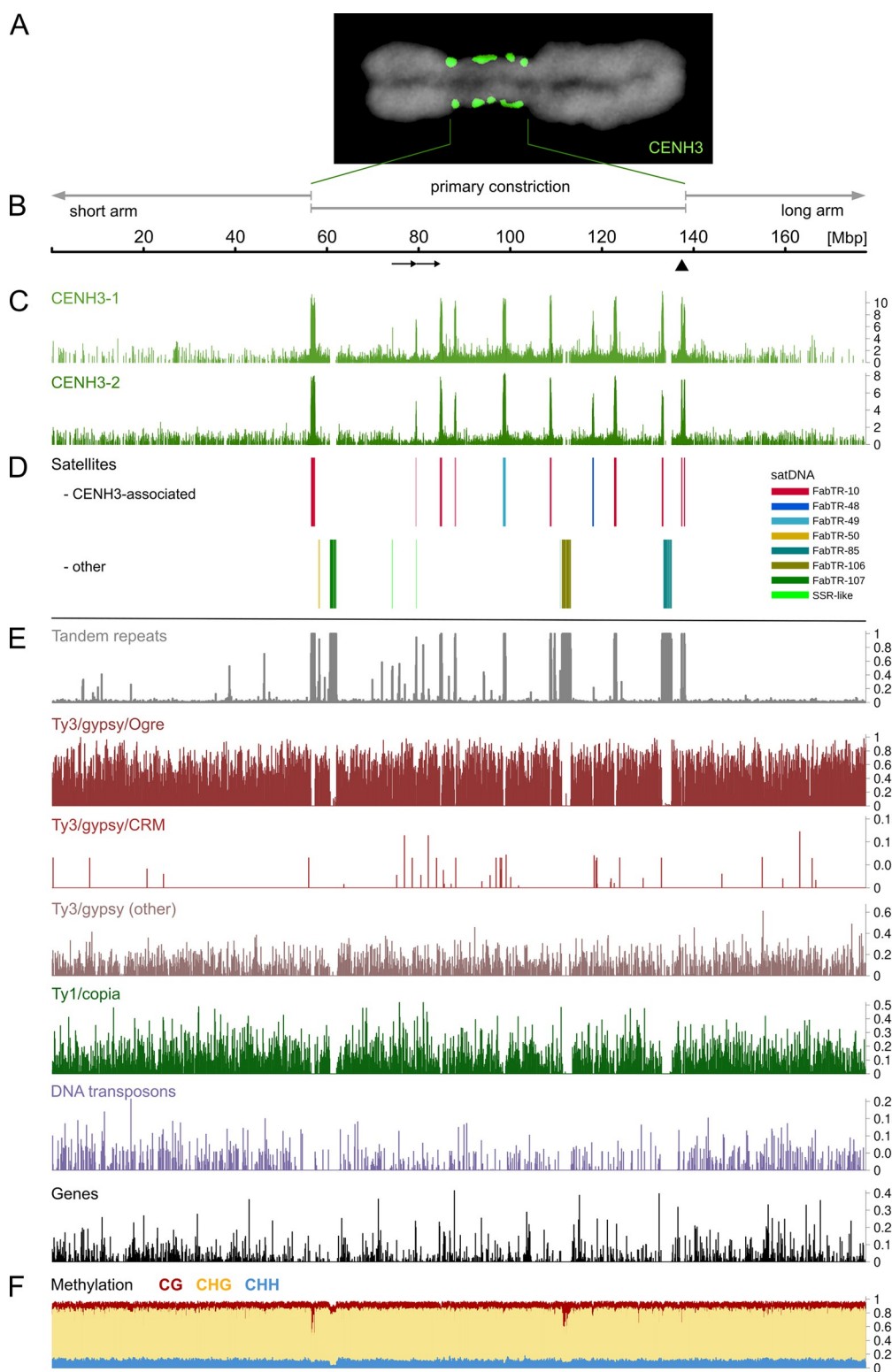

**Fig 1. Features of pea centromere 6 (CEN6). (A)** Immunolabeling of CENH3 protein (green) on metaphase chromosome 6 (counterstained with DAPI, gray). **(B)** Position of the primary constriction in the assembly. Arrows below the scale indicate the 5.2 Mb tandem duplication, and the arrowhead shows the position of a single gap in the assembly. **(C)** Distribution of CENH3 chromatin revealed by ChIP-seq experiments using anti-CENH3-1 and anti-CENH3-2 antibodies. Peaks in the graphs correspond to the statistically significant enrichment ratio of ChIP reads to

control input reads (see S2 Fig for full data analysis including experimental replicates). **(D)** Positions of large arrays of satellite repeats. Different repeat families are marked by different colors, as indicated in the legend. **(E)** Densities of different types of repetitive DNA sequences and predicted genes calculated in 100 kb windows. **(F)** Cytosine methylation profiles calculated as the ratio of methylated cytosines to all cytosines present in the sequence. Ratios were calculated separately for cytosines in three different contexts (distinguished by plot colors) and averaged for 100 kb windows.

centromere and chromosome arms, and were characterized by strong cytosine methylation in CG and CHG contexts, and mostly unmethylated CHH motifs in both regions (Figs 1F, S3A and S3D). However, there were some notable exceptions, such as a portion of the satDNA arrays, which were hypomethylated compared with the average patterns. This was most evident in the CHG motifs in FabTR-10 and FabTR-106, and in the CHH motifs in FabTR-107 (S3B and S3C Fig). In the case of FabTR-10, variation was detected among arrays located at different parts of the centromere, with arrays located near the centromere-chromosome arm junction being the most hypomethylated. Apart from these large blocks of satDNA, detailed inspection of methylation profiles along the assembly revealed smaller regions of reduced methylation, with a part of these regions overlapping with or adjacent to the genes. This finding was also reflected in the gene methylation frequency histograms, which showed hypomethylation of a substantial proportion of CG and CHG motifs, and high levels of methylation

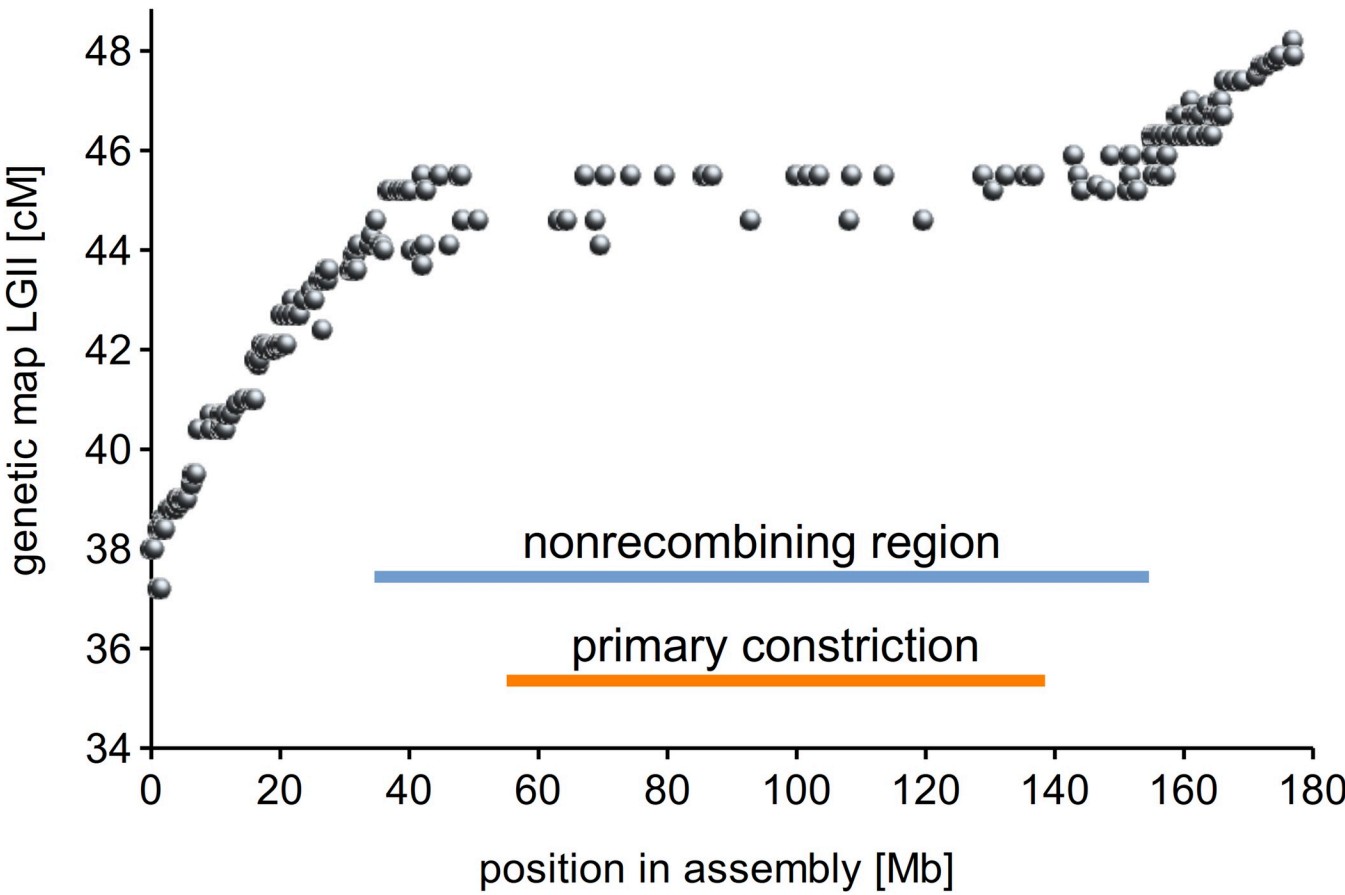

**Fig 2. The primary constriction coincides with a region of reduced meiotic recombination.** The graph shows the location of genetic marker sequences in the CEN6 assembly compared to their position in the linkage group LGII of the pea genetic map [22].

in the remaining motifs, resulting in a bimodal histograms (S3D Fig). No difference was observed between the methylation patterns of genes located within the centromere and those located in chromosome arms.

## Homogenization patterns of satDNA arrays

Similarities among monomers within individual satDNA arrays and between multiple arrays of the same repeat family are shown in Fig 3. The major satellite repeat of CEN6, FabTR-10, consisted of eight arrays (a1–a8; 230–893 kb in length), all of which were associated with CENH3 chromatin (Fig 1C and 1D). The pea genome contains two main families of FabTR-10, FabTR-10-PST-A and FabTR-10-PST-B, which differ in monomer length (459 and 1,975 bp, respectively) [20]. Although there was some variation in monomer lengths in FabTR-10 (not shown), all CEN6 arrays could be assigned to the FabTR-10-PST-A family. Additionally, dot plots of sequence similarity showed that homogenization of FabTR-10 monomers mainly occurred within individual arrays or their parts, resulting in sequence divergence between arrays at different loci (Fig 3). The only exception was the high sequence similarity between the adjacent arrays a7 and a8, indicating that these arrays originated following a recent duplication and inversion event. The orientation of monomers was uniform within each array, except in a2, which contained an inversion of a portion of the array. However, the monomers showed no preferred orientation throughout the centromere. Interestingly, the binding to CENH3 was relatively uniform across the arrays, regardless of the degree of sequence homogenization and methylation or the presence of particular sequence variants of FabTR-10 (S4 Fig).

Each of the remaining six satellites analyzed occupied a single locus in CEN6. Only two of these satellites, FabTR-48 and FabTR-49, were associated with CENH3. No major differences were observed in array homogenization patterns between CENH3-associated satellites, including FabTR-10, -48 and -49, and non-CENH3 satellites, as both groups showed patchy dot-plot patterns indicative of regions within the arrays with increased local sequence homogenization. In general, there were no trends of higher sequence homogenization at the center of the arrays. The FabTR-107 and FabTR-85 arrays showed patterns of long parallel lines, indicating segmental duplications of large portions of these arrays (Fig 3).

## Spatial arrangement of CEN6 during mitosis and interphase

We employed FISH with satDNA probes as cytogenetic landmarks to examine how the primary sequence of CEN6 transforms into the three-dimensional structure of the metapolycentromere during mitosis. The results showed that satDNA arrays associated with CENH3 domains are located along the outer periphery of the primary constriction, as required for the interaction of CENH3 chromatin with the kinetochore and mitotic spindle (Fig 4A). Each of the FabTR-48- and FabTR-49-specific probes produced a single fluorescent spot, corresponding to their respective single loci in the assembly. The probe for the major CENH3-associated repeat, FabTR-10, generated signals along the entire length of the constriction; however, the number of signals did not exactly match the number of FabTR-10 arrays in the assembly, indicating the fusion of signals from proximally positioned arrays. In contrast to the CENH3-associated repeats, the arrays of the other large satellites (FabTR-85, -106, and -107) were observed predominantly within chromatids, often near the chromosome axis, or as linear signals across the chromatid width (Fig 4B). This may be because chromatin is packed into megaloops, with CENH3 domains driven to the periphery of the constriction and the non-CENH3 chromatin constituting its interior.

Simultaneous detection of CENH3 and satellite repeats by immuno-FISH in nuclei showed that, in contrast to their multidomain structure on metaphase chromosomes, all CENH3

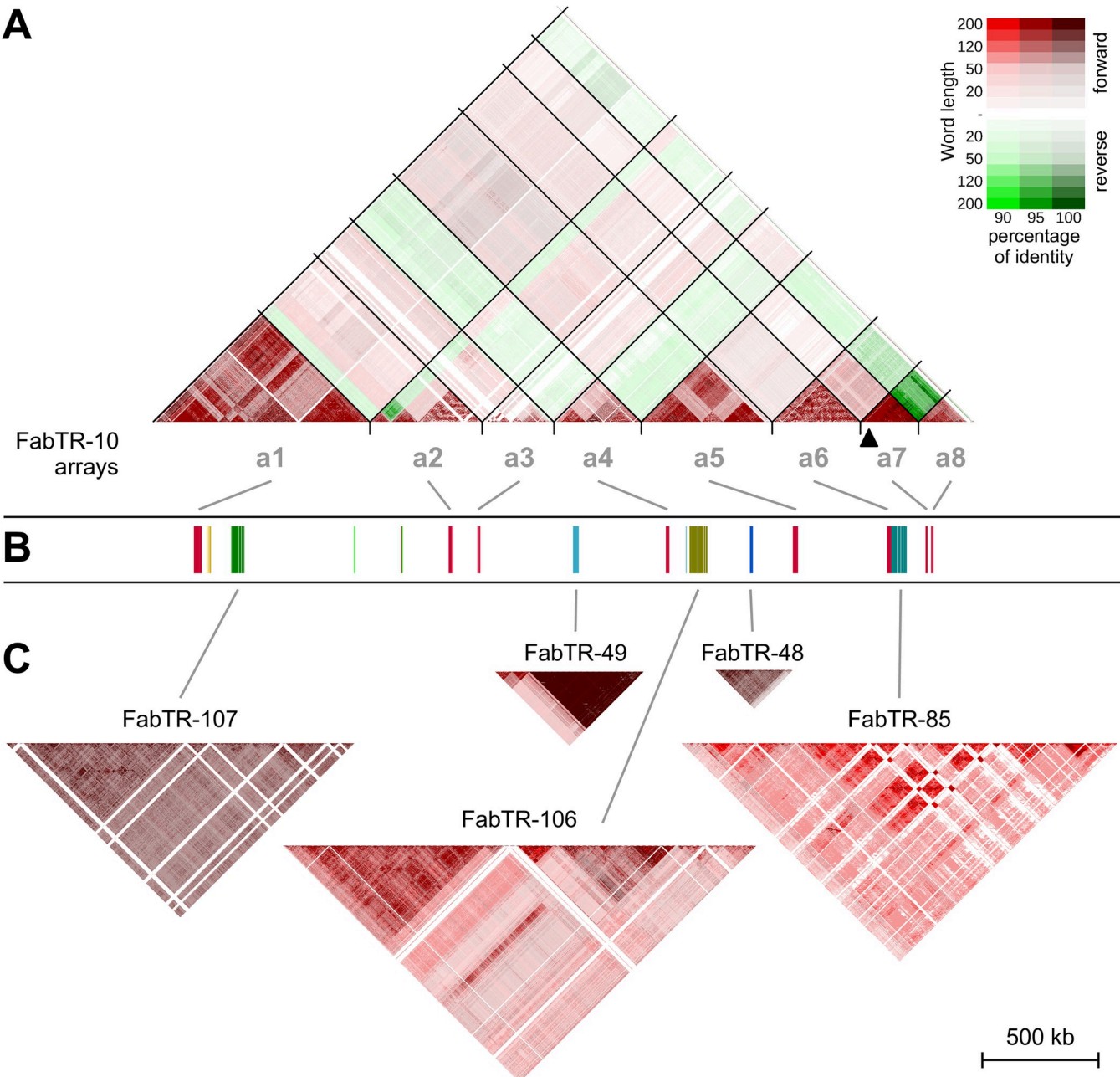

**Fig 3. Sequence homogenization patterns of satellite DNA arrays.** Nucleotide sequence similarities were visualized as similarity dot plots of k-mers of different sizes (10–200 nt). The percent identity and mutual orientation of the compared sequences are indicated by the colors shown in the legend. **(A)** Dot-plot of FabTR-10 repeats showing comparison of sequences both within and between arrays located in eight different loci (a1–a8) in CEN6 (corresponds to Fig 1D). **(B)** The schematic representation of the array positions in CEN6 (corresponds to Fig 1D). **(C)** Dot plots of the satellites present in CEN6 as single arrays. Sequence comparisons were performed only within individual arrays for these satellites. All dot plots were calculated identically and drawn to scale to account for differences in sequence homogenization and array lengths. Black arrowhead under the FabTR-10 a7 array shows the position of the gap of unknown length in the assembly. Since there are no significant sequence similarities between different FabTR families (S6 Fig), their comparisons are not plotted.

domains aggregated into a single spot per interphase chromosome, resulting in 14 CENH3 spots per nucleus (Fig 4C). Consequently, FISH signals from CENH3-associated satellites overlapped with these spots. However, FISH signals from satellite repeats not associated with

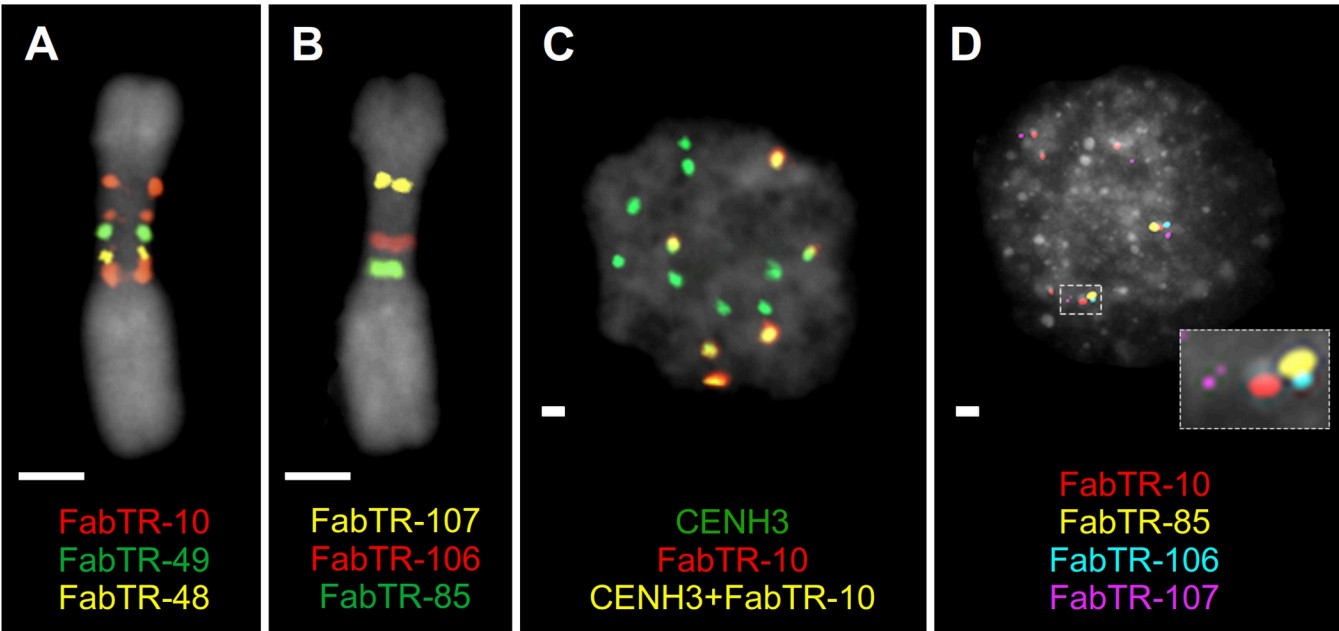

**Fig 4. Association of repeats with CENH3 determines their position on chromosomes and condensation patterns in interphase nuclei. (A-B)** Multicolor FISH detection of satellite repeats on metaphase chromosome 6. CENH3-associated satellite repeats are located along the periphery of the primary constriction **(A)**, whereas CENH3-free satellites are embedded within the constriction **(B)**. **(C-D)** Immuno-FISH detection of CENH3 protein and satellite repeats in interphase nuclei. **(C)** All CENH3 loci from each chromosome are condensed into a single spot, along with their associated satellites such as FabTR-10, resulting in 14 CENH3 signals per nucleus (2n = 14). Note that only a part of chromosomes contain FabTR-10. **(D)** CENH3-free satellites are located away from the condensed CENH3 domains of CEN6. The position of CENH3 chromatin is indicated with the FabTR-10 probe. Satellite repeats and CENH3 protein are labeled with different colors as indicated in the figures. Chromosomes and nuclei counterstained with DAPI are shown in gray. Bar = 2 μm.

CENH3, such as FabTR-85, -106, and -107, were found relatively far from the CENH3 spots, suggesting that these satellites were located on decondensed chromatin loops emanating from the densely packed CENH3 domains (Fig 4D). Overall, these experiments revealed that the spatial arrangement and condensation of different parts of the centromere sequence during the cell cycle differ, depending on their association with CENH3 chromatin.

### Elucidation of CEN6 evolution in Fabeae using oligo-FISH painting probes

Taking advantage of the CEN6 assembly, we designed a set of FISH painting probes based on oligo pools derived from single-copy regions in the assembly (Fig 5A). Two probes were designed for the primary constriction, covering either its entire length (probe PS6-C; 8,915 oligos) or a specific 3.7 Mb region within the constriction (probe PS6-C1.8; 1,800 oligos). The third probe was designed to label the regions of both the long and short arms of chromosome 6 directly adjacent to the constriction (probe PS6-A; 19,250 oligos). Despite the low average density of hybridizing oligos (0.12 oligos/kb in PS6-C and 0.26 oligos/kb in PS6-A), the probes produced relatively uniform and specific signals at their target regions (Figs 5B, 5C and S5).

To elucidate the evolution of metapolycentric chromosomes, we used the painting probes to identify the regions homoeologous to pea CEN6 in the chromosomes of selected Fabeae species (Fig 5C). In *Pisum fulvum*, the species most closely related to pea, the PS6-C probe labeled the entire constriction on one chromosome pair, with signal extending into the short arm. The signal from the PS6-A probe was correspondingly shifted, confirming that the region corresponding to the *P. sativum* CEN6 constriction short-arm junction was within the short arm of

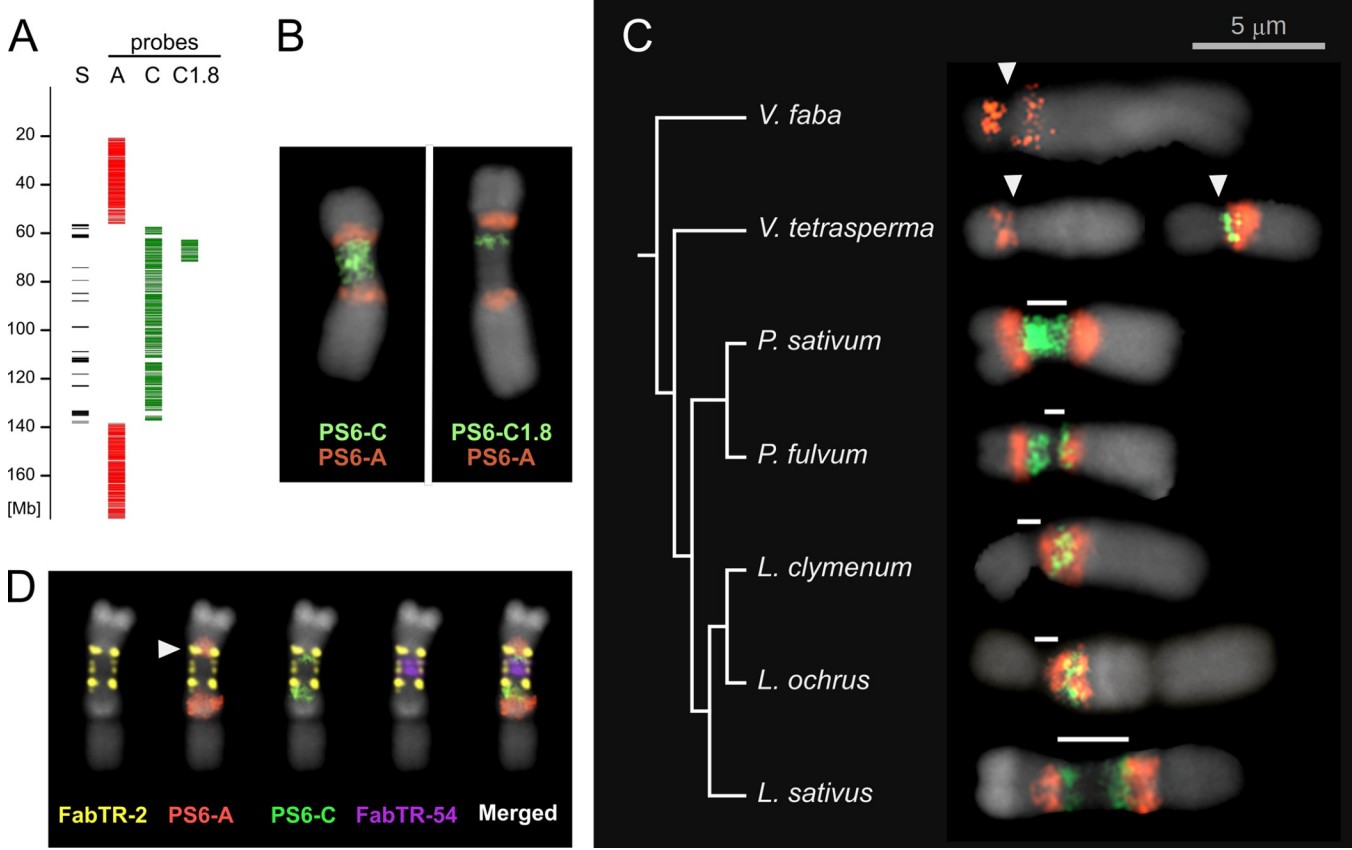

**Fig 5. CEN6 painting probes and their application for the detection of orthologous regions in related species. (A)** Positions in the assembly of oligonucleotide sequences used as FISH painting probes. Each column represents different PS6 probes. Column "S" shows the positions of satDNA arrays marking the extent of primary constriction. **(B)** Painting probes applied to *P. sativum* chromosome 6. **(C)** FISH analysis of a set of related *Fabeae* species using PS6-C (green) and PS6-A (red) probes. The phylogenetic tree was adapted from [20]. Only chromosome(s) that produced hybridization signals are shown. Primary constrictions are marked with white arrowheads (monocentric) or bars (metapolycentric chromosomes). Images of whole chromosome complements can be found in S5A Fig. **(D)** Multicolor FISH labeling of the *Lathyrus sativus* homeolog of pea chromosome 6 using PS6 painting probes as well as probes for satellite repeats FabTR-54, which fills the gap in the PS6-C signal, and FabTR-2, which is associated with CENH3 chromatin in *L. sativus* [20]. Arrowhead indicates the overlap of PS6-A and FabTR-2 signals.

*P. fulvum* chromosome 6. This observation of the shorter constriction, based on chromosomal morphology, was confirmed by CENH3 immunolabeling (S5B Fig).

We then examined representatives of the genus *Lathyrus*, which is known to share metapolycentric chromosome morphology with *Pisum*, although the size of the primary constriction varies considerably among *Lathyrus* species [7]. In *L. clymenum*, which has chromosomes with relatively short constrictions, the painting probes hybridized to a single chromosome pair, although signal intensity was weaker than that observed in *Pisum*. The probes produced the expected pattern, i.e., a single green band (PS6-C) located between two red bands (PS6-A), one on either side; however, this pattern was shifted from the centromere (as observed in *P. sativum*) into the long chromosome arm (Fig 5C). The same results were obtained for the closely related *L. ochrus*. By contrast, *L. sativus*, which has extremely elongated centromeres, showed signals that overlapped with primary constrictions on a pair of chromosomes. However, the PS6-C signal did not cover the entire constriction, leaving out the region adjacent to the short arm, and contained a large unlabeled gap within the constriction. Considering the signal of the PS6-A probe and simultaneous hybridization with the FabTR-2 probe, which marks the

positions of CENH3 chromatin in *L. sativus* [20], we concluded that the constriction on this chromosome extends into the region corresponding to the short arm of pea chromosome 6. In addition, further experiments using *L. sativus* satDNA probes developed previously [25] revealed that the gap in the PS6-C signal was caused by the amplification of the FabTR-54 repeat, which is not present in *P. sativum* (Fig 5D).

To complement our study with related Fabeae species that possess monocentric chromosomes, we applied the *P. sativum* oligo-FISH probes to two *Vicia* species: *V. tetrasperma*, which is phylogenetically closely related to the *Pisum/Lathyrus* clade, and *V. faba* (Fig 5C). The signals from the probes were more difficult to detect. In *V. faba*, the green signal (PS6-C) was completely absent, probably because it is the most distant to *P. sativum* and has a larger genome, and only weak red signals (PS6-A) were detected in the long- and short-arm regions surrounding the centromere of chromosome 3. In *V. tetrasperma*, the probes labeled centromeric regions of two chromosome pairs, indicating chromosomal rearrangements such as the reciprocal translocation of short arms.

## Discussion

Centromeres represent the final frontiers of genome projects because of their high contents of satellite repeats, which in principle are extremely difficult to assemble. However, the recent introduction of accurate long-read sequencing technologies and advanced assembly strategies has led to gapless assemblies of several complex genomes, ushering in a new era in centromere research. In plants, complete centromere assemblies have been constructed only for monocentric species to date, including maize [14,15], rice [18] and *Arabidopsis thaliana* [16,17]. In addition, high-quality assemblies of three holocentric species belonging to the *Rhynchospora* genus recently became available [26]. Here, we report the assembly of a genomic region representing yet another type of centromere organization, namely metapolycentromere, in the pea cultivar Cameor. Except a single gap in one of the satDNA arrays, the assembly is without gaps, providing the most detailed sequence information lacking in previous studies of metapolycentromeres, which mainly used cytogenetic approaches [6–8,20]. Similar to the previously reported complete assemblies of human and plant genomes, the contiguity of CEN6 assembly was enabled by the use of highly accurate long reads (PacBio HiFi), which enabled the reconstruction of most satDNA arrays, and by combining the assembly with physically localized cytogenetic markers. A unique feature of our study was the use of ultralong nanopore reads for creating manually curated scaffolds for most of the assembly, since the repetitive and complex structure of pea centromeres makes them prone to misassemblies. This makes our CEN6 assembly superior in completeness and contiguity even to the novel high-quality genome assembly of the pea cultivar ZW6 [27], which was published during preparation of this manuscript.

It has been speculated that metapolycentromeric chromosomes represent an intermediate state between monocentric and holocentric chromosomes [6,7]. Monocentric chromosomes are generally characterized by an uneven distribution of genomic features along their length, with centromeric and pericentromeric regions showing greater repetitive DNA accumulation, lower gene density, and different epigenetic profiles than the chromosome arms. By contrast, holocentric chromosomes show a more homogeneous distribution of repeats, genes, and histone modifications [26]. For example, during mitosis, histone H2A phosphorylation at Thr120 (H2AT120ph) is detected across almost the entire length of holocentric chromosomes at the outer periphery of chromatids, but is restricted to the (peri)centromeres in monocentric chromosomes [4]. In this respect, pea CEN6 is more similar to holocentromeres, as we did not detect significant differences in the distribution of genes and most repeats between the

constriction and neighboring chromosome arms. It is also noteworthy that H2AT120ph and histone H3 phosphorylation marks H3T3ph, H3S10ph, and H3S28ph have been shown to extend throughout the entire constrictions of *P. sativum* and *L. sativus* metapolycentric chromosomes [8]. On the other hand, several satDNA families accumulate in CEN6, forming long arrays, some of which are associated with CENH3. Arrays of centromeric satellites up to several megabasepairs in length are typical of monocentric chromosomes, whereas holocentric chromosomes either lack CENH3-associated satellites [28] or have them distributed as multiple short arrays across their length [26].

Although information on the long-range structure, methylation profiles, and CENH3-binding ability of centromeric satellites along the fully assembled arrays is still sparse, several common features have been reported for human alpha satellites, *Arabidopsis CEN180*, and rice CentO, including (1) the presence of chromosome-specific variants of centromeric satellites; (2) homogenization of satellite sequences within each array, often resulting in the highest similarity at the centers of arrays; (3) nonuniform binding of CENH3 along the arrays; and (4) hypomethylation of array regions associated with CENH3 [13,16–18,29]. On the other hand, CENH3 chromatin is largely restricted to the centromeric satellite arrays in humans and *Arabidopsis*, whereas this association is not as tight in rice, where most of the CENH3 is located outside the CentO arrays in some centromeres [18]. The centromeres of maize differ even more substantially; several chromosomes lack the centromeric satellite CentC, and CENH3 shows no preferential binding to CentC or to other repeats [14], suggesting that these limited observations cannot be generalized.

Our characterization of pea CEN6 provides further evidence for the diversity in plant centromeres. Instead of a single type of satellite repeat, the pea genome contains multiple distinct satellite sequences, three of which are associated with CENH3 in CEN6. Unlike the abovementioned species (*Arabidopsis*, rice, human), we observed no evidence of preferential sequence homogenization in the centers of satDNA arrays in pea, regardless of their association with CENH3. Moreover, CENH3 enrichment profiles in pea were relatively uniform along the arrays, despite their sequence variation. These observations suggest that, unlike human or *Arabidopsis* centromeres, the association of CENH3 with pea centromeric satellites is not determined by their sequence. The occurrence of multiple centromeric satellites and their rapid turnover is common in Fabeae species [20], implying that their evolution cannot be explained by the centromere drive model [3], which requires the presence of a single centromeric satellite. The question of what features make some of the pea CEN6 satellites competent for CENH3 binding remains unanswered, even considering their variation in cytosine methylation patterns (Figs 1 and S3), because we could not detect any methylation profiles that would consistently distinguish between arrays associated with CENH3 from those not associated with CENH3. For example, only some of the CENH3-binding FabTR-10 arrays were hypomethylated, but hypomethylation was also detected in some CENH3-less satellites such as FabTR-106 and FabTR-107.

One of the most intriguing questions that could be addressed, owing to the availability of the centromere assembly, is the origin and evolution of metapolycentric chromosomes. We approached this problem by developing oligo-pool FISH painting probes to identify regions orthologous to pea CEN6 in related Fabeae species. These experiments revealed the highly dynamic nature of centromere evolution in Fabeae, characterized by centromere shifts, chromosome translocations, and the expansion (and perhaps contraction) of primary constrictions. Our results support the view that the expansion of metapolycentromeres is facilitated mainly by the spreading of CENH3 chromatin from the centromere into adjacent chromosome arms, as depicted in Fig 6. However, the factor(s) triggering this process and the molecular mechanisms involved remain to be elucidated.

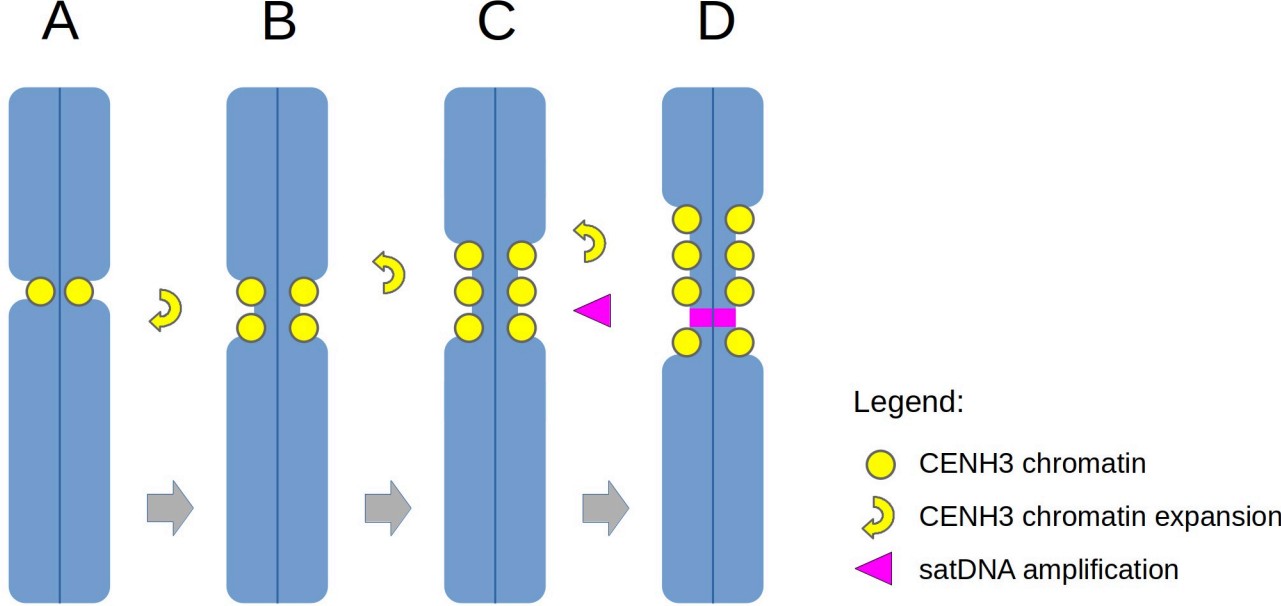

**Fig 6. A simplified model of the evolution of metapolycentric chromosomes.** A centromere of the ancestral monocentric chromosome (**A**) is expanded by the formation of new CENH3 domains near the original centromere (**B-C**) and the proliferation of satellite DNA (**D**).

Insights into the possible mechanisms involved in metapolycentrere formation could be obtained from centromere shifts reported in monocentric chromosomes (see [30] and references therein). These centromere shifts are explained either by chromosomal rearrangements such as translocations or inversions or by the repositioning of centromeric chromatin to a new location without disrupting the linear order of chromosomes [31]. Uncovering the exact mechanisms, especially in the case of centromere repositioning, depends on the availability of gapless genome assemblies of related genotypes that differ in centromere position, as defined by their CENH3 distribution. Such efforts have been initiated in the pangenome studies of maize and wheat (*Triticum aestivum*), where centromere shifts have been detected in some of the genotypes examined [15,32]. In addition, Xue and colleagues conducted a detailed investigation of the formation of a new centromere domain on rice chromosome 8 [33], and showed that the formation of this domain was triggered by the deletion of a part of the existing centromere including the CentO array. The new domain arose in a nearby genomic region, which contained increased amounts of CENH3 in the wild-type genotype. Thus, this mechanism can generate centromeres with multiple CENH3 domains, similar to metapolycentric chromosomes. However, compared with rice, the CENH3 domains in the pea CEN6 metapolycentrere are much more widely spaced and are all confined to satDNA arrays. Another mechanism, based on the mobilization of CENH3-associated centromeric satellite Tyba by Helitron elements, has been proposed to facilitate the spread of centromeric chromatin in holocentric *Rhynchospora* species [26]. However, this is unlikely to occur in pea centromeres because CENH3-associated satellites in the pea genome are organized in a few large arrays, unlike the centromeric satellites of *Rhynchospora*, which exist as a large number of scattered and much shorter loci that may be embedded in functional Helitron elements.

The only mechanism we have identified thus far that may favor the propagation of CENH3 domains in metapolycentrere and is supported by our sequence data is that of segmental

duplications, which are frequent in some plant centromeres [34]. The larger of the two segmental duplications identified in pea CEN6 originated from the region between simple sequence repeat (SSR)-like arrays and FabTR-10 arrays, and contained portions of these arrays in the duplicated sequence. Because FabTR-10 repeats are associated with CENH3, a new but relatively small (73 kb) CENH3 domain was generated 5.2 Mb downstream of the original array. However, this mechanism cannot explain the origin of other CENH3 loci because no traces of sequence duplications were detectable at these loci. Thus, segmental duplication could be just one of several synergistic forces driving the evolution of metapolycentric chromosomes.

To gain further insight into the rapid and divergent evolution of centromeres in Fabeae, several research directions are conceivable. A new improved version of the whole-genome sequence of pea cv. Cameor, based on the sequence data and methods described in this study, is currently under construction and is expected to provide near-complete assemblies of the remaining six centromeres. Sequence comparison of these centromeres with CEN6 (described here) will enable the identification of common features of evolutionary or functional significance. This approach will be further strengthened by the inclusion of the highly contiguous genome assemblies of related species, such as *L. sativus* (metapolycentric) and *V. faba* (monocentric), which are also in progress [35]. In addition to the investigation of centromere properties, these assemblies should also be used for the comparative analysis of kinetochore genes to reveal any differences in kinetochore composition among species with different centromere organization. The rationale for this approach stems from the finding that the transition to holocentricity in some groups of organisms is accompanied by the loss or multiplication of CENH3 or other kinetochore genes [36–38], similar to the duplication and diversification of CENH3 genes in *Pisum* and *Lathyrus* [7].

## Materials and methods

### Genomic DNA preparation and sequencing

High molecular weight (HMW) DNA was prepared from the nuclei extracted, and subsequently purified, from the young leaves of pea (*Pisum sativum* L. cv. 'Cameor') seedlings, as described previously [25]. The quality of DNA preparations was checked using field inversion gel electrophoresis (FIGE) to ensure that the DNA fragment size was >100 kb. Then, 3–40 μg of input HMW DNA was subjected to 20 runs of nanopore sequencing on the MinION sequencer (Oxford Nanopore Technologies) using the following library preparation kits, according to the manufacturer's instructions: SQK-LSK109 (13 runs), SQK-LSK110 (1 run), SQK-RAD004 (3 runs), and SQK-ULK001 (3 runs). Raw nanopore reads were basecalled using Oxford Nanopore basecaller Guppy (ver. 3.6.0 and 4.5.4). Quality-filtering of the resulting FastQ reads and their conversion to FASTA format were performed with BBDuk (part of BBTools, https://jgi.doe.gov/data-and-tools/bbtools/) using the quality cutoff parameter maq = 8. Reads shorter than 30 kb were discarded. PacBio HiFi reads were generated from the same input HMW DNA by DNA Sequencing Center of the Brigham Young University (UT, USA) using four SMRT Cells on a PacBio Sequel II instrument by running the Circular Consensus Sequencing (CCS) protocol for 30 h.

### CEN6 scaffolding and assembly

A fraction of the ultralong nanopore reads (>160 kb) was used to create scaffolds covering most of the assembled region. The scaffolding process was initiated by identifying "seed" nanopore reads, which contained sequences of genetic markers located in the nonrecombining region of linkage group LGII in the pea high-density genetic map [22]. A portion of these marker sequences were also detected on metaphase chromosomes with the highly sensitive FISH protocol, which

was used to determine their exact physical location (S1 Fig). Additional physically localized seed reads were derived from the edges of the arrays of satellite repeats, FabTR-48, -49, and -50, which were previously shown to be specific to CEN6 [6,20]. Next, the seed reads were extended in both 5' and 3' directions by searching the database of ultralong reads using BLASTN [39] and minimap2 [40] for similarities with their 60 kb terminal regions. The identified read overlaps were verified by sequence similarity dot plots automatically generated using Gepard [41] and by manual inspection, ensuring that the extending read sequence was confirmed by at least one other overlapping read. Eventually, if the extending or confirming reads could not be obtained from the longest fraction, collections of reads shorter than 160 kb were searched. The verified extending reads were then merged with the seed reads to form initial scaffolds. This process was then iterated using the end regions of scaffolds as queries in the next round of similarity searches and extensions until two adjacent scaffolds were merged. Alternatively, the extensions were stopped when the scaffolds reached highly homogenized regions of some satellite repeats that prevented the reliable selection of overlapping reads, because of the relatively high error rate of nanopore reads. On the other hand, higher sequence variation and the presence of mobile element insertions in satellite arrays allowed them to be reliably scaffolded with long nanopore reads.

The assembly of HiFi reads was performed using Hifiasm assembler [42] version 0.16.1, with default parameters. Alternatively, HiCanu [43] version 2.1.1 was used with the options "genomeSize = 4.2G useGrid = false -pacbio-hifi". Contigs from the HiFi assemblies were used to cover the regions that were not scaffolded using nanopore reads (mostly within the long arm of chromosome 6, S1 Fig). The HiFi contigs were also used to fill gaps in the nanopore scaffolds corresponding to satDNA arrays. With the exception of the a7 array of satellite FabTR-10, which was not fully represented in any HiFi contig, all satDNA arrays were fully assembled and were therefore used to represent these regions in the assembly. Finally, the assembly was polished with HiFi reads using Racon version 1.4.20 [44].

## Assembly annotation

Annotation of repetitive sequences was performed using a combination of different tools available on the RepeatExplorer Galaxy Server (https://repeatexplorer-elixir.cerit-sc.cz/). Transposable element sequences encoding conserved protein domains were identified based on their similarities to the REXdb database [45] using DANTE (https://github.com/kavonrtep/dante). Full-length LTR-retrotransposon sequences were annotated using the DANTE_LTR tool (https://github.com/kavonrtep/dante_ltr), which combines the results of DANTE with similarity- and structure-based identification of LTR-retrotransposon signatures such as LTRs, primer binding sites (PBSs), and target site duplications (TSDs). The identified full-length LTR-retrotransposons were also used to create a reference database for similarity-based annotation of repeats in the assembly. The database was also enriched with consensus sequences of repeats obtained from the RepeatExplorer analysis of Fabeae genomes [46] and with a collection of Fabeae satDNA sequences compiled on the basis of our previous studies [20,25,46]. In parallel with similarity-based detection, tandem repeats were identified, based on their genomic organization, with Tandem Repeats Finder ver. 4.09 [47] using the parameters "2 5 7 80 10 500 2000". The output of the search was parsed and converted to GFF format using TRAP [48].

Gene annotation was performed by launching FINDER [49] on the CEN6 assembly supplemented with unscaffolded HiFi contigs representative of the remaining parts of the genome. Briefly, 30 RNA-seq libraries [50,51] were mapped to the assembly by STAR, and assembled with psi-class [52]. Next, the mapped data were processed by braker2 [53] to perform a *de novo* annotation of genes. To improve the quality of annotation, Ryūtō [54] was run twice on the mapping results, once for the stranded library and the second time for the unstranded

library. The results of Ryūtō and psi-class were combined using Mikado [55] to obtain a high-quality (HQ) annotation dataset. A low-quality (LQ) dataset was built by filtering braker2 results as follows. First, genes overlapping a repeat annotation were removed. Then, only the genes with at least one hit in the eggNOG protein database were retained. Functional annotation of these genes was performed using TRAPID with the PLAZA Dicots 4.0 database.

## CENH3 ChIP-seq analysis

ChIP experiments were performed with native chromatin as described previously [6], using custom antibodies that specifically recognize one of the two variants of pea CENH3 proteins. DNA fragments were purified from the immunoprecipitated samples, and the corresponding control samples (Input; digested chromatin not subjected to immunoprecipitation) were sequenced on the Illumina platform (Admera Health, NJ, USA) in paired-end, 150 bp mode. Duplicate experiments, including independent chromatin preparations, were performed for each CENH3 variant using either one antibody (P23 for CENH3-2) or two different antibodies (P22 and P43 for CENH3-1); both anti-CENH3-1 antibodies were raised against an identical peptide in rabbit (P22) and chicken (P43). The antibodies were targeted to the N-terminal tail, which is the most variable part of the CENH3 proteins, resulting in only 35% similarity of the peptide used to raise the CENH3-2 antibody P23 to the CENH3-1 sequence, and 43% similarity of the peptide used to raise CENH3-1 antibodies P22 and P43 to the CENH3-2 sequence [6–8]. The resulting reads were quality-filtered and trimmed using Trimmomatic [56] (minimum allowed length = 100 nt), yielding 122–211 million reads per sample, which were mapped onto the assembly using Bowtie 2 version 2.4.2 [57], with options -p 64 -U. Subsequent analysis was performed on full output from Bowtie2 program and on output where all multimapped reads were filtered out. Filtering of multimapped reads was performed using Sambamba version 0.8.1 [58] with options "-F [XS] = = null and not unmapped and not duplicate". Regions with statistically significant ChIP/Input enrichment ratio were identified by comparing ChIP and Input mapped reads using the epic2 program [59], with the parameter "—bin-size 200". Alternative identification of enrichment was performed using MACS2 [60] version 2.1.1.20160309, with default settings. The ChIP/Input ratio was calculated for plotting purposes using bamCompare (version 3.5.1) from the deepTools package [61]. The program was run with the parameter "–binSize 200" to calculate the log2 ratio for the 200 nt window size. The resulting data were plotted using the rtracklayer package of R [62].

## Methylation analysis

Cytosine methylation was analyzed in all three contexts (CG, CHG, and CHH) by detecting the frequency of 5-methyl cytosine (5mC) in nanopore reads, which were aligned to the CEN6 assembly using DeepSignal-plant ver. 0.1.4 [24] with the model "model.dp2.CNN.arabnrice2-1_120m_R9.4plus_tem.bn13_sn16.both_bilstm.epoch6.ckpt". Prior to the analysis, nanopore reads were rebasecalled using the latest version of Guppy (ver. 6.0.1) and resquiggled using Tombo ver. 1.5.1. Methylation frequencies were calculated for each cytosine position in the assembly, based on the number of methylated and methyl-free cytosines detected in the aligned nanopore reads. The methylation analysis pipeline was run on a Linux server equipped with 126 GB RAM, 24 CPUs, and the NVIDIA GeForce GTX 3060 graphics card.

## Bioinformatics analysis

Unless stated otherwise, all data handling and bioinformatic analyses were implemented using custom Python, Perl, and R scripts, and executed on a Linux-based server equipped with 256 GB RAM and 48 CPUs.

## Centromere painting probe design and FISH

The painting probes were designed on the basis of unique 45 nt oligos, which were selected from specific regions of the CEN6 assembly using the Chorus2 program [63]. The probes were then synthesized by Daicel Arbor Biosciences (Ann Arbor, MI, USA) either as myTags Custom Labeled Probes (PS6-C, labeled with Alexa Fluor 488; PS6-A, labeled with ROX) or as myTags Custom Immortal Probe PS6-C1.8, which was subsequently labeled with biotin-16-dUTP, as described previously [64]. The satDNA-based probes were either synthesized as an oligo-pool probe (oPools Oligo Pools, IDT) or cloned and labeled with Alexa Fluor 568 or 488 (Thermo Fisher Scientific, Waltham, MA, USA) via nick translation [65]. The cloned probes for single-copy expressed sequence tag (EST)-based genetic markers were labeled with Alexa Fluor 488 or Alexa Fluor 568 (Thermo Fisher Scientific, Waltham, MA, USA) using nick translation.

Mitotic chromosomes used for cytogenetic analyses were prepared from synchronized root apical meristems [7]. After cell cycle synchronization, chromosome preparations were obtained using different protocols, depending on their end use: single-copy FISH targets and centromere painting probes [66], satDNA-based probes [20], or CENH3 immunolabeling [20,67]. Pachytene chromosomes were extracted from anthers as described previously [68], with some modifications. Flower buds (3–5 mm in size) were collected, fixed in Carnoy's solution (3:1 ethanol: acetic acid) overnight at room temperature, and then transferred to 70% ethanol and incubated at 4˚C until needed for further analysis. After rinsing with distilled water for 5 min, the flower buds were washed twice with 1× citrate buffer for 5 min each time. Finally, the flower buds were dissected, and the anthers were removed and placed on a microscope slide in a drop of 60% acetic acid, where they were squashed under a coverslip.

FISH using painting probes and satDNA-based probes was performed as described previously [69], with hybridization and washing temperatures adjusted to account for the probe AT/CG content. Hybridization stringency was modified to allow for 10% mismatches (when hybridized to *P. sativum* chromosomes) or 20–30% mismatches (when hybridized to the chromosome preparations of other species). When performing FISH using painting probes, 3–10 pmol of the probe was used per slide; post-hybridization washes were conducted in 0.1× SSC instead of 50% formamide/2× SSC; and the biotin-labeled PS6-C1.8 probe was detected using streptavidin-Alexa Fluor 488 (Jackson Immunoresearch). FISH using satDNA oligo-pool probes was performed according to the method described previously [70], with some modifications. Briefly, after rinsing in 2× SSC, the chromosome preparations were fixed in 45% acetic acid for 4 min, postfixed in 2× SSC containing 4% formaldehyde for 10 min, and washed in 2× SSC for 10 min after each fixation. Following dehydration in an ethanol series (50%, 70%, and 96%), 20 µl of the hybridization mix (50% [v/v] formamide, 10% dextran sulfate in 2× SSC, and 30–100 pmol of the oligo-pool probe) was applied to each slide with chromosome preparations, which was then incubated at 84˚C for 3 min to induce DNA denaturation. After 20 h of hybridization, all washes were performed at 37˚C. Single-copy FISH was performed as described previously [66].

To perform multicolor FISH, up to two rounds of rehybridization were performed. To remove the previously hybridized probes, the slides were washed at room temperature in 4× SSC/0.2% Tween 20 for at least 30 min and twice in 2× SSC for 5 min, then in 50% formamide/2× SSC for 10 min at 55˚C, and finally in 2× SSC for 10 min at room temperature. Samples were postfixed before proceeding with the next hybridization. Immunolabeling, combined with FISH, was conducted as described previously [20].

## Supporting information

**S1 Fig. Assembly construction and verification using genetically and physically localized markers.** The nanopore "seed" reads used to initiate CEN6 scaffolding were selected based on

the presence of sequences of genetic markers from the nonrecombining region of linkage group LGII or the sequences of CEN6-specific satellite repeats. (**A**) The positions of genetic marker sequences in the assembly (x-axis) compared with their positions on the genetic map. Markers highlighted in green were physically localized on chromosomes (panel F). (**B**) The position of the primary constriction in the assembly. Arrows below the scale indicate the 5.2 Mb tandem duplication, and the arrowhead indicates the position of a single gap in the assembly. (**C**) Positions of the satDNA arrays, with the three CEN6-specific families marked with asterisks. (**D**) Regions of the assembly that were scaffolded with nanopore reads or constructed from HiFi contigs are shown by horizontal bars. Dots mark gaps in nanopore scaffolds corresponding to satDNA arrays that were filled using HiFi contigs. (**E-F**) Examples of assembly verification using FISH. (**E**) Localization of selected satellite repeats on pachytene chromosomes. Note that smaller FabTR-10 signals are not visible due to the short exposure time. (**F**) Sequences of genetic markers (green) detected on metaphase chromosome 6 using the highly-sensitive single-copy FISH protocol. Satellite PisTR-B (red) was used to discriminate chromosomes within the pea karyotype.
(PDF)

**S2 Fig. Localization of centromeric chromatin by CENH3 ChIP-seq.** Duplicate experiments were performed for each CENH3 gene variant using either two different antibodies (P22 and P43 for CENH3-1) or one antibody (P23 for CENH3-2). The number of reads mapped onto the assembly was presented either as a ratio of ChIP-seq reads to genomic (input DNA) reads (lanes "ratio") or as regions of significant ChIP-seq enrichment identified with the epic2 and macs2 programs. (**A,B**) Mapping of reads onto the assembly either in multilocus mode (**A**) or single-mapping mode (**B**). In (**A**), multiple mappings of repetitive reads were allowed. In (**B**), only the reads with unique hits were mapped, and repetitive reads were discarded.
(PDF)

**S3 Fig. DNA methylation profile of CEN6.** Per-base cytosine methylation frequencies in three sequence contexts known in plants (CpG, CHG, CHH) were obtained by analyzing Oxford Nanopore reads aligned to the assembly using DeepSignal-plant (Ni et al., 2021) [24]. (**A**) The plots show the fraction of aligned nanopore reads, in which cytosine was methylated at a given position. The total number of aligned nanopore reads is indicated in the "coverage" plot. The distribution of CENH3 chromatin and annotations of the major families of satDNA are shown for comparison with the methylation profiles. (**B,C**) Detailed examples of hypomethylated regions. Hypomethylated arrays of satDNA are marked with asterisks. (**B**) Sequence at the short-arm constriction junction contains CHG-hypomethylated FabTR-10, whereas the array of the same repeat within the constriction has a normal methylation level (**C**, marked with "x"). Short hypomethylated islands are best seen in the gene-rich region marked in (**C**). The CENH3 ChIP-seq track shows enrichment peaks identified by the epic2 program using multi-mapped reads generated with the P22 antibody. (**D**) Per-base methylation frequency distributions within specific regions or sequence types. Distributions were calculated for the entire primary constriction ("CEN") and chromosome arm ("arms") sequences as well as for specific satellite repeats and genes. Gene sequences occurring in the centromere (CEN) and chromosome arms were analyzed separately. Red arrowheads mark the position of peaks corresponding to hypomethylated genes.
(PDF)

**S4 Fig. CENH3 ChIP-seq and methylation profiles of FabTR-10 arrays.** The data shown represent zoomed-in sections of the graphs shown in S2 and S3 Figs corresponding to loci with FabTR-10 arrays. The positions of the arrays are indicated by gray bars below the graphs

and are complemented by sequence homogenization dot plots (compiled from Fig 1).
(PDF)

**S5 Fig. FISH with CEN6 painting probes.** (**A**) Chromosome complements of selected Fabeae species hybridized with PS6-C (green) and PS6-A (red) painting probes. Bar = 10 μm. (**B**) Hybridization pattern of CEN6 painting probes on chromosome 6 of *Pisum fulvum*. *Left panel*: extent of the primary constriction (white bar), as revealed by the immunolabeling of CENH3 and the FISH detection of PisTR-B repeats, showing that PisTR-B is located just above the CENH3 signals. *Right panel*: combined FISH detection using the painting probes together with the PisTR-B probe, which was used as a reference for the end of the constriction and shows that the green PS6-C probe extends into the short arm.
(PDF)

**S6 Fig. Monomer lengths and sequence similarities between satellite repeat families present in CEN6.** The dot-plot shows all-to-all sequence comparison of consensus monomer sequences representing individual satDNA families. Similarities were scored within a sliding window of 100 bp and dots or lines were drawn when at least 50 matching bases were detected. A single monomer copy is compared for all families, except for families with short monomers, where three (FabTR-85) and ten (FabTR-107) concatenated monomers were used for comparison. The lengths of the consensus monomers are given in parentheses.
(PDF)

## Acknowledgments

We thank V. Tetourová and J. Látalová for technical assistance, and D. Beránková and E. Hřibová (Institute of Experimental Botany, Olomouc, Czech Republic) for help with labeling myTags Custom Immortal Probe. We also thank the ELIXIR CZ Research Infrastructure Project (LM2018131) for providing computing and data-storage facilities.

## Author Contributions

**Conceptualization:** Jiří Macas, Judith Burstin.

**Data curation:** Jiří Macas, Jonathan Kreplak, Petr Novák.

**Funding acquisition:** Jiří Macas.

**Investigation:** Jiří Macas, Laura Ávila Robledillo, Jonathan Kreplak, Petr Novák, Iva Vrbová, Pavel Neumann.

**Methodology:** Andrea Koblížková.

**Resources:** Andrea Koblížková, Judith Burstin.

**Software:** Petr Novák.

**Visualization:** Laura Ávila Robledillo, Petr Novák, Iva Vrbová, Pavel Neumann.

**Writing – original draft:** Jiří Macas.

**Writing – review & editing:** Jiří Macas, Laura Ávila Robledillo, Jonathan Kreplak, Petr Novák, Andrea Koblížková, Iva Vrbová, Judith Burstin, Pavel Neumann.

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
