## [Decision Letter · Decision Letter 0]

15 Dec 2022

Dear Dr Macas,

Thank you very much for submitting your Research Article entitled 'Assembly of the 81.6 Mb centromere of pea chromosome 6 elucidates the structure and evolution of metapolycentric chromosomes' to PLOS Genetics.

The manuscript was fully evaluated at the editorial level and by independent peer reviewers. The reviewers appreciated the data reported in this manuscript and are encouraging publication. However, they also raised some important points that we would like you to consider to improve the manuscript. Based on the reviews, we will not be able to accept this version of the manuscript, but we would be willing to review a revised version.

If you decide to revise the manuscript for further consideration at PLOS Genetics, please aim to resubmit within the next 60 days, unless it will take extra time to address the concerns of the reviewers, in which case we would appreciate an expected resubmission date by email to plosgenetics@plos.org.

We are sorry that we cannot be more positive about your manuscript at this stage. Please do not hesitate to contact us if you have any concerns or questions.

Yours sincerely,

Claudia Köhler

Section Editor

PLOS Genetics

Reviewer's Responses to Questions

**Comments to the Authors:**

Reviewer #1: The authors deciphered the largest-ever analyzed centromere, the so-called “metapolycentromere” of the pea chromosome 6. This 81.6 Mb-long centromere was analyzed to understand the organization of such an unusual centromere type and to get insight into the molecular and evolutionary mechanisms leading to centromere-type variations. Generally, two types of centromeres exist, mono- and holocentric chromosomes, but how holocentromere evolved is still unclear. Notably, the metapolycentromere shares several features of a holocentromere. Therefore metapolycentric chromosomes provide a unique chance to study the changes associated with the transition between different centromere organizations.

I enjoyed reading this manuscript because all conclusions are based on conclusive data, and the findings are new and exciting. I have only a few comments which are listed below.

Page 2, line 40

Remove “plant”, because the same applies to non-plant species

Page 3, line 84.

Please be more precise. The sat DNA interacts with CENH3-containing nucleosomes.

Page 5, line 117

The title is not understandable for a non-specialist. I would suggest (or similar) “Assembly of the polymetacentromere of pea chromosome 6”

Page 6, line 161

The observation that both types of CENH3-antibodies immunoprecipitated the same sequences is interesting. Please could you indicate the similarity between both CENH3 variants?

Page 6, line 164.

‘To emphasize the different centromere structure, I would suggest adding “extended”. It would read: “The extended primary constriction showed….”

Page 7, line 230

Just a note, the same observation regarding the linear signals of centromere repeat and linear signals across chromatids of non-centromeric satDNA was reported for some Rhynchospora species, see: http://www.ncbi.nlm.nih.gov/pubmed/27645892

Page 10, line 313

Please change to:” histone H2A phosphorylation at Thr120 ((H2AT120ph) is detected across almost the entire length of holocentric chromosomes at the outer periphery of chromatids, but is restricted….

About the suggested model metapolycentromere formation, a request

Please, would it be possible to prepare a schemata which would summarize the potential process of metapolycentromere formation? I think that future readers would appreciate such a summary.

Figures

The size bars are missing in most chromosome/interphase pictures. Please add.

Reviewer #2: The authors describe a comprehensive assembly and annotation of a 177.6 Mb region of pea (Pisum sativum) chromosome 6, including the 81.6 Mb centromere region (CEN6) and adjacent chromosome arms, with the intent of interpreting how metapolycentromeres evolved. They provide evidence for an evolution via the expansion of centromeric chromatin into neighboring chromosomal regions with accumulation of novel satellite repeats. The assembly and annotation is of very high quality, and the manuscript may stand alone on these merits. The assembly revealed a clear evolutionary interpretation of centromere dynamics in metapolycentric chromosomes. They report that pea metapolycentromeres are strikingly composed by different families of satellite repeats subdivided into distinct centromere domains in contrast to most commonly found single-domain repeat-based monocentromeres. Furthermore, the authors nicely describe and interpret the genomic and cytological data with well-prepared figures.

The CENH3-ChIP-seq experiments were well designed and performed, including the necessary controls and replicates.

The repeat annotation and both in silico and in situ analysis of centromere chromatin is impressive.

The oligo-painting FISH analysis presents a very clever strategy to study the evolution and expansion of metapolycentromeres.

Some comments follow below:

I am aware of a new development of the Hifiasm assembler which now integrates HiFi reads and Ultralong Nanopore reads. I wonder if the authors tried that assembly strategy on their data already and if it would make sense to compare the assemblies using this strategy, in order to remove the single gap present in CEN6.

The centromere effect on recombination rates of metapolycentromere compared to typical monocentromeres is remarkable. I am surprised the authors give so little detail and do not discuss these important findings. Is the centromere effect similar to monocentromeres? Is there a gradient reduction of CO rates in pericentromeric regions? Or it is rather an abrupt transition between hot and cold CO regions?

This study is interesting primarily because of the unique centromere structures of Pea chromosomes. The findings that metaplycentromere evolved rather by extension of the original monocentromere is interesting. It is unclear what sequence features if any Pea centromeric repeats have that facilitate CENH3 deposition in comparison to non-centromeric ones. Clearly, they show similar high order organizations.

So, although much of what is presented on metapolycentromere organization was already known from previous papers from the same group, this manuscript is an excellent genome analysis of nearly T2T Pea chromosome 6. The centromere analysis included in the study is very well done and helps to understand the evolution of metapolycentromeres and could potentially be a way for the evolution of repeat-based holocentromeres. The paper would have been more complete had they carried ChIP-seq using more histone marks as well to get a more comprehensive perspective of the overall epigenetic regulation of metapolycentromeres.

Reviewer #3: This impressive study reports characterization of metapolycentric centromeres from Pisum, which is a structure intermediate between mono- and holo-centric. Metapolycentric species show a large primary cytological constriction accounting for up to a third of the chromosome. This is a novel centromere organisation and the authors provide a detailed molecular portrait, which represents a new contribution to the field. However, there are a number of improvements I would like to see before acceptance.

The authors use long read ONT and HiFi data to generate an assembly of pea CEN6. A semi manual approach to assembly is used, in addition to automated methods, eg HiFiasm. It would be interesting to see coverage analysis of the ONT and HiFi reads aligned back to final assembly.

The paper focuses on CEN6, as it is one of the largest in the genome – however, if automated assembly has been performed I wasn’t clear why the other centromeres have not been reported?

An interesting picture with respect to centromere tandem repeats is observed. There are multiple centromere located tandem repeat families, some of which are CENH3 occupied. Generally the relationships between the reported repeats, if any, are not well described. Simple histograms of repeat lengths would be useful to see. Which of the repeats are more closely related, and does this correlate with CENH3 occupancy? One tool I would recommend be used is TRASH https://github.com/vlothec/TRASH which will allow more precise definition of tandem repeat families. It would also be interesting to define the occurrence of satellite higher order repeats, which is functionality included with this tool.

A further tool that may be nice to use is StainedGlass https://github.com/mrvollger/StainedGlass This would allow a single identity heat map to be made over CEN6 that could be presented with the array specific dotplots shown in Fig. 2. A StainedGlass map would provide a nice overview of sequence identity across the entire CEN6 region.

The immunocytology data in Fig 4A-B is very nice, as us Fig 3C which indicates repeat family turnover between species.

I also liked Fig. S1, particularly the clear correlation with cM data – I think this is worth considering for a main figure?

More analysis on the gene content in the polymetacentromere region, compared to the rest of the arms would be good. It seems there are abundant genes between the CENH3 peaks. Are there are striking GO differences? Or differences in RNA-seq levels? Are these genes more heterochromatic (ie higher DNA methylation)?

The transposon analysis is interesting – were any TEs found within the satellite arrays, or in areas of CENH3 enrichment?

The CHG methylation looks very high >80% along the chromosome – in other plants CHG would be lower eg 30-40%. The ONT based calls from Deepsignal-plant are trained on Arabidopsis and rice, which may not be the best model for pea? Due to the very high CHG levels it would be worth to compare to regular BS-seq data to confirm this finding.

Interestingly, Fig S3 (I think this also should be in the main figures) shows a very interesting difference between two TR-10 arrays that are both CENH3 occupied, but only one shows a CHG drop, as observed in Arabidopsis (Naish et al). On line 147 – please provide information on the ChIP-seq read lengths. Due to the repetitiveness of these sequences, what controls or efforts were made to confirm that short read alignments were accurate in these regions? In the methods I note that multi mapped reads were dropped – could this create biases? Which ChIP-seq data is shown in Fig. S3B-C - is this unique mappers, or all mapped reads?

On line 144 – please provide more detail on the annotation methods used for these distinct sequence classes.

Line 171 – Im unclear what ‘no significant effect’ means here – clearly the sequences are present and so I would say they do have a significant effect – perhaps the authors mean that they are a relatively small amount of the sequence?

Line 360 – ‘CEN180’ should be italics.

**Have all data underlying the figures and results presented in the manuscript been provided?**

Reviewer #1: None

Reviewer #2: Yes

Reviewer #3: Yes

PLOS authors have the option to publish the peer review history of their article (what does this mean?). If published, this will include your full peer review and any attached files.

Reviewer #1: No

Reviewer #2: No

Reviewer #3: No

---

## [Decision Letter · Decision Letter 1]

17 Jan 2023

Dear Dr Macas,

Thank you very much for submitting your Research Article entitled 'Assembly of the 81.6 Mb centromere of pea chromosome 6 elucidates the structure and evolution of metapolycentric chromosomes' to PLOS Genetics.

As you will see, two of the reviewers are satisfied with the revisions, but reviewer 3 has some justified requests about additional information to be included in the manuscript. We would therefore ask you to include the requested information where possible. Since there are no figure restrictions, you can also include relevant information into the main figures.

Yours sincerely,

Claudia Köhler

Section Editor

PLOS Genetics

Reviewer's Responses to Questions

**Comments to the Authors:**

Reviewer #1: I'm satisfied with the revision of the manuscript and with the newly prepared figure 6 (proposed model).

Reviewer #2: The authors have addressed all my comments and concerns.

Reviewer #3: The authors have made an adequate attempt to respond to my comments, but more documentation would have been beneficial for a couple of points.

They mention having run StainedGlass, but that it yielded a blank plot. This is surprising, as they already show dotplots for the repeat arrays that should certainly be seen in a StainedGlass output. Provision of the plot from this approach in the revision document would be helpful to see even if it doesn't figure in the main manuscript.

I still think inclusion of the cM data in the main figures would be beneficial. I wasn't aware that this journal has strict limitations on figures and my recommendation to the editor would be to allow space for this important data if there is a limitation.

The response to my question about LTR insertions in the satellite arrays is also very cursory and lacks any significant detail.

Overall this is an excellent paper that will be of interest to the field and certainly worthy or publication in Plos genetics.

**Have all data underlying the figures and results presented in the manuscript been provided?**

Reviewer #1: Yes

Reviewer #2: Yes

Reviewer #3: Yes

PLOS authors have the option to publish the peer review history of their article (what does this mean?). If published, this will include your full peer review and any attached files.

Reviewer #1: No

Reviewer #2: No

Reviewer #3: **Yes: **Ian Henderson

---

## [Editor Report · Decision Letter 2]

23 Jan 2023

Dear Dr Macas,

We are pleased to inform you that your manuscript entitled "Assembly of the 81.6 Mb centromere of pea chromosome 6 elucidates the structure and evolution of metapolycentric chromosomes" has been editorially accepted for publication in PLOS Genetics. Congratulations!

Yours sincerely,

Claudia Köhler

Section Editor

PLOS Genetics

Claudia Köhler

Section Editor

PLOS Genetics

Comments from the reviewers (if applicable):

**Data Deposition**

http://datadryad.org/submit?journalID=pgenetics&manu=PGENETICS-D-22-01340R2

**Press Queries**

---

## [Editor Report · Acceptance letter]

31 Jan 2023

PGENETICS-D-22-01340R2 

Assembly of the 81.6 Mb centromere of pea chromosome 6 elucidates the structure and evolution of metapolycentric chromosomes 

Dear Dr Macas, 

We are pleased to inform you that your manuscript entitled "Assembly of the 81.6 Mb centromere of pea chromosome 6 elucidates the structure and evolution of metapolycentric chromosomes" has been formally accepted for publication in PLOS Genetics! Your manuscript is now with our production department and you will be notified of the publication date in due course.

With kind regards,

Anita Estes

PLOS Genetics

On behalf of:
